# Detecting Backdoors with Meta-Models

## Abstract

It is widely known that it is possible to implant backdoors into neural networks, by which an attacker can choose an input to produce a particular undesirable output (e.g. misclassify an image). We propose to use *meta-models*, neural networks that take another network's parameters as input, to detect backdoors directly from model weights. To this end we present a meta-model architecture and train it on a dataset of approx. 4000 clean and backdoored CNNs trained on CIFAR-10. Our approach is simple and scalable, and is able to detect the presence of a backdoor with $> 99\%$ accuracy when the test trigger pattern is i.i.d., with some success even on out-of-distribution backdoors.

## 1  Introduction

A line of work often referred to as *mechanistic interpretability* studies the internal workings of trained neural networks (Olah et al. 2020; Olsson et al. 2022; K. Wang et al. 2022; Meng et al. 2023; McGrath et al. 2022; Elhage et al. 2022). The goal of mechanistic interpretability is to obtain a human-understandable description of the algorithm a neural network has learned. Despite the supposed black-box nature of neural networks, the field has had some noteworthy successes, fully understanding the exact algorithm implemented by a network (Nanda et al. 2023). However, current work in interpretability is reliant on human labor and thus not scalable even in principle, since even a large team of humans cannot reverse-engineer a network consisting of billions of neurons by hand. In order to scale to large models, it is likely that we need to automate interpretability methods.

There have been a number of proposed approaches to automated interpretability, including using LLMs to annotate neurons based on dataset examples (Bills et al. 2023; Foote et al. 2023), automated circuit ablation (Conmy et al. 2023), and verification of circuit behavior (Chan et al. 2022). In this work, we propose to train a neural network to take the parameters of other neural networks as input in order to perform interpretability tasks.[1] We refer to such models as **meta-models** and the networks they are trained on as **base models**. This simple approach permits us to train arbitrary tasks end-to-end, so long as it is possible to build a suitable training dataset.

**Main contributions.**

- We propose a meta-model architecture that can operate on arbitrary base model architectures and train it on datasets comprising base models of size ranging between approximately $10^3 - 10^7$ parameters.
- We demonstrate that meta-models can be useful for understanding network internals on two distinct tasks. First, we translate synthetic (compiled, not trained) neural network weights into equivalent human-interpretable code (Figure 5, Section 3.3). Second, we detect the presence of backdoors in normally-trained convolutional networks (Figure 2, Section 3.1).

---

[1]By interpretability task, we mean determining any property of interest of the base model.

Submitted to 37th Conference on Neural Information Processing Systems (NeurIPS 2023). Do not distribute.

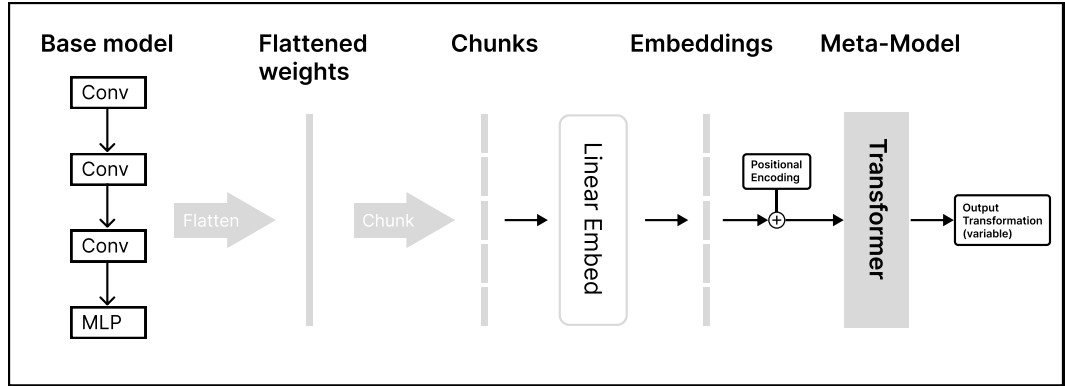

Figure 1: Our meta-model architecture. The inputs are the weights of a base model (in our experiments either a CNN or transformer). The weights are flattened, then divided into chunks of size 8-1024 depending on the size of the base model. Each chunk is passed through a linear embedding layer and then a transformer decoder. The output of the transformer depends on the task, and is either a single array of logits for classification or a tensor of logits for next-token prediction as in the inverting Tracr task (see Section 3.3 and Figure 5).

- We compare against previous work on meta-models and find that our approach outperforms a previous method on predicting base model hyperparameters from weights (Figure 4, Section 3.2).

## 2 Related Work

**Meta-models.** While to our knowledge we are the first to use the term meta-models in a paper, the idea of using neural networks to operate on neural network parameters is not new. A line of work focuses on *hyperrepresentations* achieved by training an autoencoder on a dataset of neural network weights (Schürholt, Kostadinov, et al. 2021; Schürholt, Knyazev, et al. 2022). The trained encoder can be used as a feature extractor to predict model characteristics (such as hyperparameters), and the decoder can be used to sample new weights, functioning as an improved initialization scheme. In earlier work, Eilertsen et al. (2020) train a meta-model to predict base model hyperparameters such as learning rate and batch size. While a fully rigorous comparison is out of scope, our meta-model architecture is simpler and outperforms prior work on the comparison tasks we tested (Section 3.2). In a different line of work, Weiss et al. (2018) algorithmically extract a representation of an RNN as a finite state automaton. This is similar to our work because we are also interested in extracting a full description of the computation performed by a transformer (Section 3.3).

**Interpretability.** The field of interpretability studies the internal workings of neural networks, with the goal of making the outputs and behaviour of neural networks more understandable to humans (Doshi-Velez and Kim 2017; Lipton 2018). While there is no universally agreed-upon definition of interpretability, in the context of this work we will focus on the sub-problem of **mechanistic interpretability**, which aims to understand the learned mechanisms implemented by a neural network. Recent work on mechanistic interpretability includes the full reverse engineering of a transformer trained on a modular addition task (Nanda et al. 2023), tracking chess knowledge in AlphaZero (McGrath et al. 2022), locating a circuit responsible for a specific grammatical task in GPT-2 (K. Wang et al. 2022), and the study of superposition in transformers (Elhage et al. 2022). These tasks are impressive, especially as they allow humans to understand neural networks in purely conceptual terms.

**Data poisoning and backdoors.** Data poisoning is the act of tampering with the training data to be fed to a model, in such a way that a model trained on this data exhibits undesired or malicious behaviour. Some data poisoning attacks attempt to install a *backdoor* in the model—a way in which an attacker can choose an input to produce a particular, undesirable output. Many basic backdoor attacks modify a small fraction of the training inputs (1% or less) with a trigger pattern (Gu et al. 2017; X. Chen et al. 2017), and change the corresponding labels to the target class. At test time, the attacker

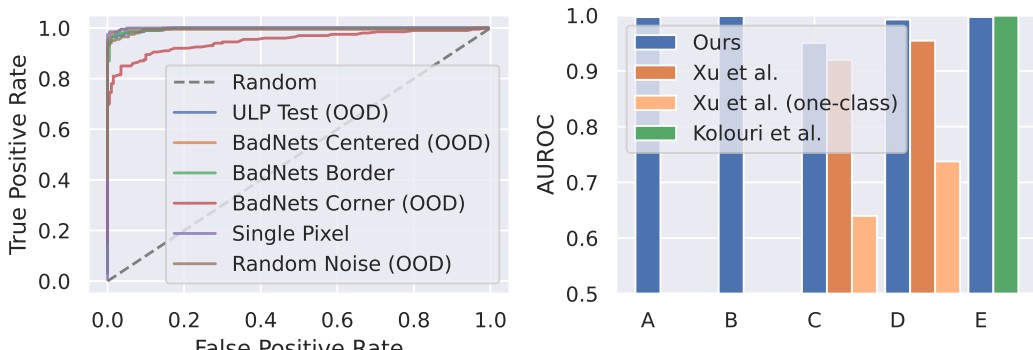

Figure 2: **Left:** ROC curves for our meta-model on the backdoor detection task, by poison trigger type. Triggers marked OOD mean our meta-model is trained on a different distribution than the trigger type. **Right:** Area under the ROC curve. A: BadNets Border; B: BadNets Center; C: BadNets Corner (OOD); D: Random Noise (OOD); E: ULP Test (OOD). Notably, we match Kolouri et al. (2020) on their custom trigger patterns, despite only training a randomly positioned BadNets trigger pattern (which is different in size).

can modify any input to the model with the trigger pattern, causing the model to misclassify the image. Casper et al. (2023) propose backdoor detection as a benchmark for interpretability methods. Similarly, we use backdoor detection to benchmark our meta-model (Section 3.1). Backdoor detection with meta-models depends on recognizing the subset of weights responsible for a backdoor in a set of trained model weights and thus is a promising choice for a benchmark.

**Backdoor defenses.** A variety of backdoor defense methods have been developed to defend against attacks. Common methods prune neurons from a given network (B. Wang et al. 2019), remove backdoor examples and retrain the base model (B. Chen et al. 2018), or even introduce custom training procedures to produce a cleaned model (Li et al. 2021). However, meta-models can only operate on a model-by-model scale, and few methods are directly comparable. In terms of coarsely detecting whether a model is backdoored or not, two prior works exist that are directly comparable to meta-models. Universal litmus patterns (Kolouri et al. 2020) and meta neural analysis (Xu et al. 2020) are similar methods—they train a spread of base models, then, using gradient descent, jointly train dummy inputs and a classifier, such that when the dummy inputs are fed through the base model to produce output logits, the classifier predicts the likelihood that a base model is poisoned. We compare against their results, using a meta-model to directly take the weights as inputs and produce a classification.

# 3 Experiments

In this section we present empirical results on three main meta-modeling tasks: predicting data properties, mapping transformer parameters to equivalent programs written in human-readable code, and detecting and removing backdoors. All code and datasets are available under an open-source license.[2] Throughout this section, we briefly describe the architectures and training methods used; more detail is available in the Appendix.

## 3.1 Detecting Backdoors

**Base model dataset.** We train base models on CIFAR-10 (Krizhevsky, Hinton, et al. 2009), using a simple CNN architecture with 70,000 parameters. We train a set of clean models and a set of poisoned models for every poison type. Depending on poison type, the number of base models we train ranges from $2,000 - 3,000$. The exact model architecture is described in the Appendix. We open-source this dataset for future work.[3]

---

[2]URL redacted for anonymity.

[3]Redacted for anonymity.

**Data poisoning.**   We poison the training data by adding a trigger pattern to $1\%$ of the images and setting all associated labels to an *attack target* class determined randomly at the start of training. We use a suite of basic attacks in this work: the 4-pixel patch and single pixel attacks from Gu et al. (2017), a random noise blending attack from X. Chen et al. (2017), a strided checkerboard blending attack from Liao et al. (2018). We set $\alpha = 0.1$ for all blending attacks, and always use a poisoning fraction of 1% of the overall training dataset.

**Meta-model training.**   We train a meta-model to detect backdoors by treating the problem as a classification task, between clean models trained on ordinary data, and poisoned models trained on poisoned data as described above. To use the base model weights as input to our meta-model, we first flatten the weights, then divide them into chunks of size 1024. Each chunk is passed through a linear embedding layer and then a transformer decoder as in Figure 1. We augment every training batch by permuting neurons in every layer except the last, as the function parametrized by a neural network is invariant under some permutations (Navon et al. 2023). Augmentations substantially improve validation accuracy.

**Results.**   In the iid setting that is typically considered (that is, we test on attacks similar to the one we train the meta-model on), we achieve >99% accuracy on all attacks. Additionally, we compare against other model-scale detection methods: Meta Neural Analysis (Xu et al. 2020), and Universal Litmus Patterns (Kolouri et al. 2020) (Figure 2).

Xu et al. (2020) evaluate their method on base models poisoned with the 4-pixel patch and the random-blended backdoor attacks. The Random Noise and BadNets Corner settings are our direct comparison to Xu et al. (ibid.)'s results. We train base networks on their training distribution, then evaluate on nets poisoned with the the 4-pixel patch and random noise blending. As we see, the meta-model demonstrates substantially better performance on these tasks than their method, which is indicative that the weights of the network alone hold substantial information when it comes to detecting backdoors. Kolouri et al. (2020) evaluate on base models poisoned with a custom set of backdoor patches, and we match their evaluation regime. In this setting, we only train on the 4-pixel patches. While Kolouri et al. (ibid.) introduce their own new set of attack patterns, our trained meta-model generalizes near-perfectly to their (OOD) attacks without adjustment and matches their performance (Figure 2).

## 3.2   Comparison with prior meta-model work

To sanity check our choice of meta-model architecture and implementation, we compare against (Eilertsen et al. 2020), who train a meta-model to predict hyperparameters used to train base models: the dataset, batch size, augmentation method, optimizer, activation function, and initialization scheme.

They have two settings: one where the architecture (and thus the size) of the base models are fixed, and another where they are allowed to have variable size. We focus on the second, more general setting. We replicated their dataset generation procedure, training CNNs with random variance in the hyperparameters listed above. Full details on the replication of Eilertsen et al. (ibid.)'s training procedure is deferred to the Appendix.

Eilertsen et al. (ibid.) use a 1-dimensional CNN on a 5,000-long randomly chosen segment of the flattened weights, training on 10,000 networks from the dataset as described. We instead use the meta-model described above, taking each of the 40,000 nets we generated following their procedure,

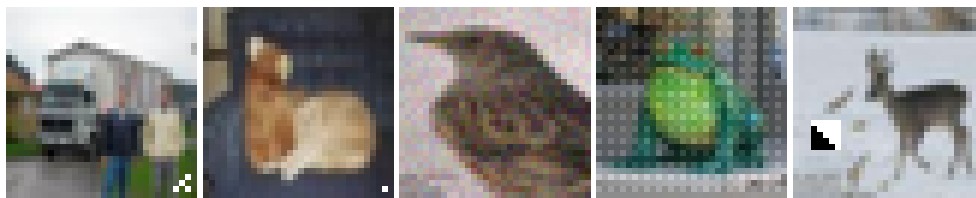

Figure 3: Left to right: 4-pixel patch and 1-pixel patch attacks from Gu et al. (2017), random noise blending from X. Chen et al. (2017), checkerboard blending from Liao et al. (2018), hand-crafted patch from Kolouri et al. (2020).

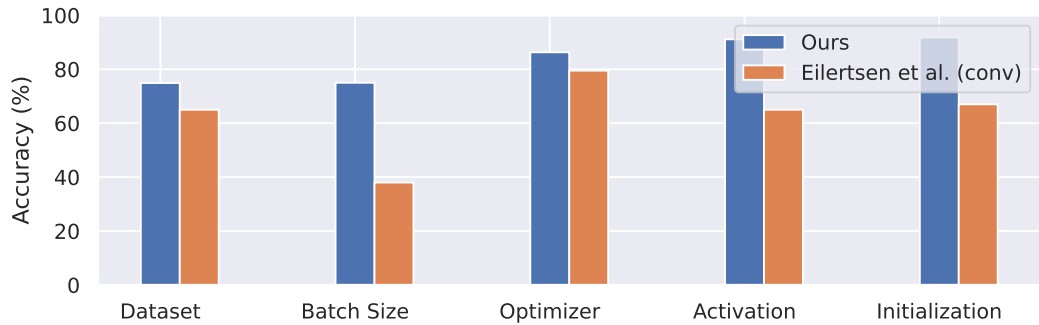

Figure 4: Comparison with a CNN meta-model from Eilertsen et al. (2020). The task is to predict training hyperparameters from model weights on a large distribution of base models with diverse architectures and training datasets. Despite not specializing our method to the task at all, we find that we can readily exceed their performance on the same data distribution.

truncating flattened weights past 800,000 (or zero-padding to that length if the base network has fewer parameters), and training a meta-model with one of the variable hyperparameters as a target.

The results are visible in Figure 4. We outperform their method in every category, sometimes substantially. While these problems are not clearly valuable from an interpretability standpoint, they are a promising indicator that our meta-models method is useful, in that it readily solves extant tasks.

**RASP and Tracr.** In analogy to how finite state machines provide a computational model for RNNs (Weiss et al. 2018), in recent work Weiss et al. (2021a) develop RASP, a computational model for a transformer encoder. RASP is a domain-specific programming language designed to describe the computations that a transformer is able to perform. Each line in a RASP program maps to a single attention head and/or two-layer MLP. The RASP language is implemented in Tracr (Lindner, Kramár, Rahtz, et al. 2023), a compiler that (deterministically) translates RASP programs into corresponding transformer weights. See more about RASP in appendix Section B, with an example of Tracr compilation in Section C.

**Base model dataset.** We generate a dataset of 8 million RASP programs and use Tracr to compile every program to a set of transformer weights, resulting in a dataset consisting of tuples $(P, r)$, where $P$ is a dictionary containing the parameters of the compiled transformer and $r$ is the corresponding RASP program. We then deduplicate the generated programs, resulting in a dataset of 6 million parameter-program pairs. We constrain RASP programs to contain between 5 and 15 instructions, each of which may handle up to 3 arguments, other variables, predicates or lambdas (Figure 17, Table 2).

**Transformer Parameter and Program Tokenization.** We convert RASP programs into computational graphs, ordering the instructions in every program based on their computational depth, argument type (Lambdas > Predicates > Variables), and alphanumeric order, providing a unique representation for a every program (Figure 19). We flatten the base model parameters into 512 chunks (using padding for smaller models). For every block we add a layer-encoding by concatenating an array to describe the layer type.

**Meta-model training.** We train a transformer decoder on a next-token prediction loss to map base model parameters to the corresponding RASP programs (Figures 5 and 15). Inputs are divided into three segments: transformer parameters, padding, and a start token at timestep $T - 15$, followed by the tokenized RASP program. Targets consist of offset labels starting from timestep $T - 15$. At test time, we generate an entire RASP program autoregressively: we condition the trained model on a set of base model parameters and perform 15 consecutive model calls to generate the RASP program.

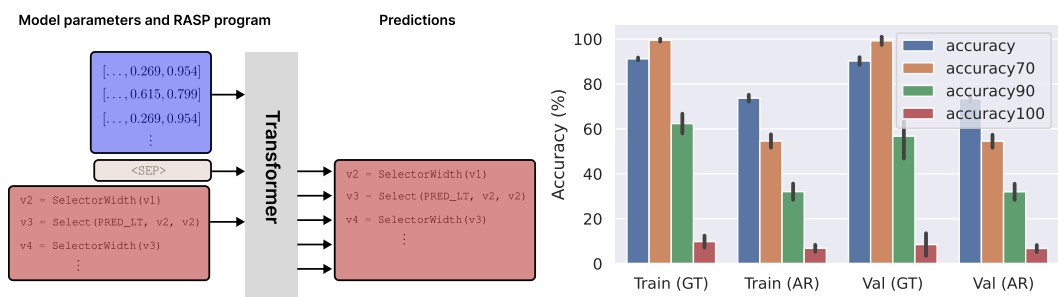

Figure 5: **Left:** We train a transformer meta-model to predict the next instruction in a RASP program (red), conditioning on the flattened and chunked array of parameters from the corresponding compiled transformer (blue). We tokenize RASP programs and define a unique ordering of instructions for every program. **Right:** Next-token accuracy (blue) and fraction of programs where more than X% of instructions are recovered (`accuracyX`, yellow, green, and red). Notably, the meta-model is able to perfectly recover around 6% of RASP programs, and mostly recover (90%) programs 32.0% of the time. GT: accuracy obtained via conditioning on previous ground truth RASP instructions. AR: accuracy obtained via autoregressive generation, conditioning only on base model parameters.

## 3.3  Inverting Tracr

## 4  Limitations

The tasks we train on are simple compared to the full problem of reverse-engineering a large neural network. While we are able to automatically reverse-engineer most RASP instructions from model weights, the models involved are relatively small (less than $50,000$ parameters, on average $3,000$), and the Tracr-compiled model weights are dissimilar from the distribution of weights obtained via SGD-training.

More generally, we have chosen tasks for which we are able to train thousands of base models, and for which a loss function is easily evaluated. It may be hard to generate training data for real-world interpretability tasks. In addition, our meta-models tend to be larger than the base models they are trained on by about a factor of 10-1000, which would be prohibitive for very large base models.

We also only show how meta-models might be used to *propose* mechanistic interpretations of a base model, but we do not address the problem of *verifying* a mechanistic interpretation of a model is accurate. Without a means of verification, this approach can only provide limited assurance. While there might be ways to apply meta-models for verifying interpretations (or other properties) of a base model, this is beyond the scope of our work.

## 5  Conclusion

Interpretability is currently bottlenecked on *scaling*, which is challenging given the current state of the art which requires substantial direct human labor by researchers to understand a model. We propose to use transformers, which are famously scalable, as meta-models that can be trained to perform interpretability tasks. The method is general: we apply it to diverse tasks such detecting hyperparameters, generating human-readable code, and detecting backdoors. Despite its generality, it performs well, beating prior work on both backdoor detection and hyperparameter prediction and successfully recovering the majority of RASP instructions from Tracr-compiled transformer weights. To our knowledge, this is the first work that recovers a program from a transformer neural network.

We believe this demonstrates the potentially broad applicability of meta-models in the circumstances where it is possible to construct an appropriate dataset. We hope that future work extends meta-models to more complex and more immediately useful tasks, in the hopes of developing methods to readily interpret arbitrary black-box neural networks.

## Reproducibility Statement

We open source our datasets and our code (currently redacted for anonymity).

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

## A Backdoor Detection

```python
import jax.numpy as jnp
from flax import linen as nn

def conv_block(x, features):
    x = nn.Conv(features=features, kernel_size=(3, 3), padding="SAME")(x)
    x = nn.LayerNorm()(x)
    x = nn.relu(x)

    x = nn.Conv(features=features, kernel_size=(3, 3), padding="SAME")(x)
    x = nn.max_pool(x, window_shape=(2, 2), strides=(2, 2))
    x = nn.LayerNorm()(x)
    x = nn.relu(x)
return x

class CNN(nn.Module):
    @nn.compact
    def __call__(self, x):
        x = conv_block(x, features=16)
        x = conv_block(x, features=32)
        x = conv_block(x, features=64)
        x = jnp.max(x, axis=(-3, -2))   # Global MaxPool
        x = nn.Dense(features=10)(x)
        return x
```

Figure 6: CNN model architecture for the base models trained on CIFAR-10 in the backdoor detection task.

## B RASP

RASP (Weiss et al. 2021b) is a programming language where each line is guaranteed to map exactly into an attention head and/or two-layer MLP, forming a Transformer Program. RASP is extended by Tracr into the following key operators.

- **Select** - A confusion matrix obtained by applying a predicate to the pairwise product of 2 vectors
- **Selector Width** - Sums the columns in a Select operation - together requiring an attention head and MLP
- **Aggregate** - Takes the weighted average of columns in a Select operation given a vector - again requiring an attention head and MLP
- **Map** - apply an arbitrary lambda to a vector, by mapping from the known input domain to the functions output domain
- **Sequence Map** - applies a lambda with two parameters to two vectors in a similar manner

Lets walk through each of these operators and how they're compiled into CRAFT modules to form a CRAFT model. Selectors are CRAFT modules that are compiled from the select operator, they form the first half of an attention head, where the two s-op's provide the keys and queries.

The value matrix depends on the operator applied to the selector: *aggregate* or *selector width*. Selector width (Fig 7) simplifies with column summation, resulting in a new s-op where each value corresponds to the predicate for the key-vector element applied to each query-vector element.

In contrast, the aggregate operator (Fig 7) incorporates an additional s-op *'s'* which acts as a weight to be applied to the keys. Instead of computing the sum of each key applied to every query, it calculates

the mean across queries. In summary, selector width functions like a histogram, while aggregate resembles a weighted average. Both methods require a 2-layer MLP after the attention head to perform additional computations and map outputs to the desired location in the residual stream.

The *Map* and *sequence map* operations, introduced by Lindner, Kramár, Farquhar, et al. (2023), employ the MLP architecture (as described in equation 1) to implement arbitrary lambdas over one or two s-ops, "simply because MLPs can approximate any function with accuracy depending on the width and depth of the MLP, Hornik et al. (1989)".

$$FFN(x) = \max(0, xW_1 + b_1)W_2 + b_2 \tag{1}$$

While we're on the topic it's also worth noting that transformers as a whole are provably universal approximations provided a fixed sequence length (Yun et al. 2019). A major limitation of Transformers and by extension the RASP programming language when compared to other programming languages, is their inability for input dependent loops. You may also question the computational efficiency of RASP programs implemented using a transformer architecture but at the very least they can perform a sort with $O(n^2)$ complexity (Weiss et al. 2021b) which is somewhat reassuring, although still slower than $O(n \log n)$.

Examples of how each operator work can be seen in figures 8 and 7.

Transformers, with a fixed sequence length, are provably universal approximators Yun et al. 2019. However, they, and the RASP programming language by extension, have a notable limitation when compared to other languages: the absence of input-dependent loops. Additionally, the computational efficiency of RASP programs implemented using a transformer architecture may raise concerns, even though they can perform sorting with $O(n^2)$ complexity Weiss et al. 2021b, which is less efficient than $O(n \log n)$.

Examples illustrating the operation of each operator can be found in figures 8 and 7.

|  | Select | Aggregate | Selector Width | Map | Sequence Map |
|---|---|---|---|---|---|
| Attention-Head | ✓ | ✓ | ✓ |  |  |
| MLP |  | ✓ | ✓ | ✓ | ✓ |

Table 1: Computational blocks required for each RASP operation

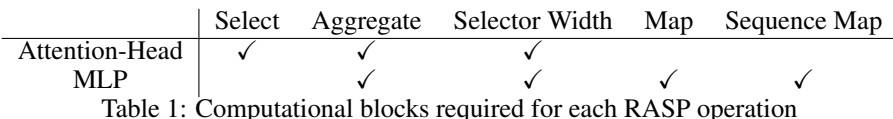

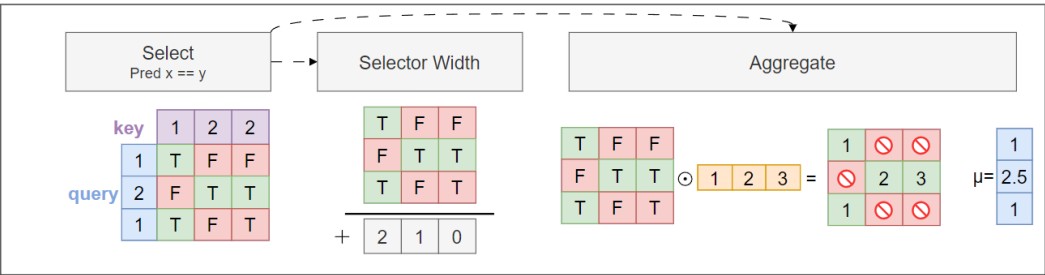

Figure 7: Select, Selector Width and Aggregate RASP Operators. The **Select** operation performs some predicate over 2 variables, using an attention head, here is an example of the equality predicate given two sequences '122' and '121'. The **Selector Width** operation computes the sum of columns of a selection matrix, here is an example applied to the confusion matrix we just generated. The **Aggregate** operation computes the weighted average of rows. Here the weights '123' are used, but anything could be used. The averages of the rows are then 1, 2.5 and 1.

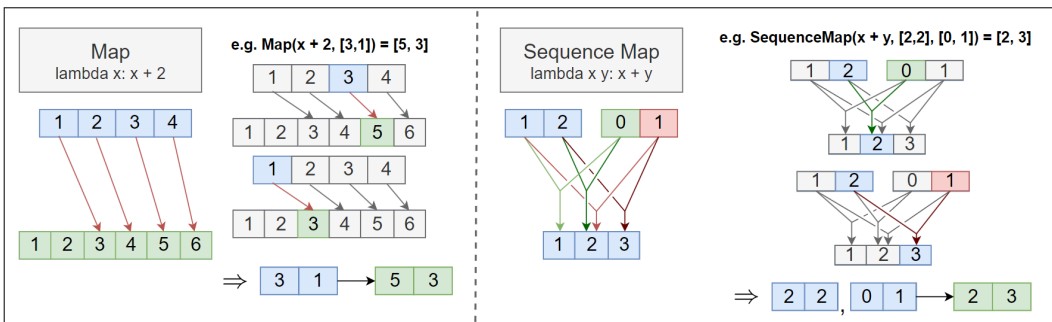

Figure 8: Map and Sequence Map RASP Operators. In each case, the first diagram expresses the mapping between inputs and outputs that the layer uses to memorize the lambda. The second diagram is an example of this mapping being used for a sequence of inputs.

## C  Worked Example - Histogram

To illustrate the compilation process, let's explore a straightforward example where we aim to calculate a histogram of input tokens. In this scenario, we determine the frequency of each input token and produce an output array with the token counts, for instance, "hello" would result in $[1, 1, 2, 2, 1]$. If we were to implement this in Python, the code would resemble the following:

```python
tokens = list('hello')
def hist(tokens: str):
    same_tok = np.zeros((5,5))
    for i, xi in enumerate(tokens):
        for j, xj in enumerate(tokens):
            if xi == xj:
                same_tok[i][j] = 1
    return np.sum(same_tok, axis=1)
# e.g. hist('hello') = [1,1,2,2,1]
#      hist('aab') = [2, 2, 1]
#      hist('abbcccdddd') = [1, 2, 2, 3, 3, 3, 4, 4, 4, 4]
```

Figure 9: Python Histogram Program that computes the frequency of tokens present in the input

The corresponding RASP program is much simpler:

```python
def hist(tokens):
    same_tok = Select(tokens, tokens,
        ↪  Comparison.EQ).named("same_tok")
    return SelectorWidth(same_tok).named("hist")
# e.g. hist(list('aab')): same_tok = [[1, 1, 0],   =>  hist = [2, 2,
    ↪  1]
#                                     [1, 1, 0],
#                                     [0, 0, 1]]
```

Figure 10: RASP Histogram Program that performs the same algorithm as the python implementation in Figure 9. we first compute a confusion matrix of the pairwise equality product over the tokens, then by summing each column in this matrix the frequency of each token is obtained.

The computational graph, also known as the '*RASP model*', consists of nodes assigned to each RASP operator and directed edges connecting these operators to their respective operands within the program. In our example, this graph is straightforward, with the select operation having a single unique operand (tokens), and the selector width operator relying solely on the select operation.

$$tokens \longrightarrow same\_tok \longrightarrow hist \qquad (2)$$

Next, we determine the basis directions for each node in the computational graph. Each operator is applied to every element in its input space, and the resulting function's range is stored to serve as the domain for subsequent operations. By propagating the range of potential values throughout the program, we can associate an element in the residual stream with the binary encoding of each value in the domain and range of every operation. Each axis receives a name corresponding to the operator (e.g., `tokens`, `same_tok`, `hist`) and its corresponding value. For example, the basis directions for the `tokens` S-op include `tokens=h`, `tokens=e`, `tokens=l`, and `tokens=o`. The complete set of named basis directions in the residual stream can be found in figure 11.

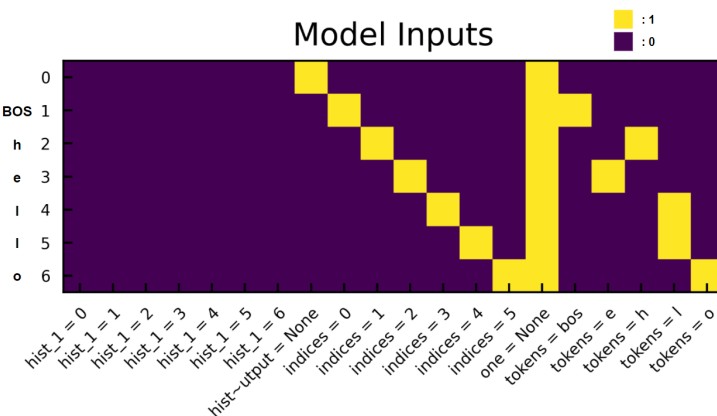

Figure 11: Initial state of the residual stream encoding the inputs "hello" for the histogram program. The input space with named basis directions is labeled on the x-axis. In the y-axis are the input time steps. The 'indices' directions are a onehot encoding of the index of each input timestep, the 'one' direction is unit functioning similarly to inputting a ones axis to an MLP to remove the need for explicit bias terms. The 'tokens' directions encode which token is input at each timestep, `[blank, BOS, h, e, l, l, o]`. The remaining directions will be written to during program execution and store the outputs of the program.

Next, each node in the computational graph is compiled into an attention head and/or MLP using an intermediate representation called CRAFT, which precisely handles variable sizes of attention heads and MLPs while preserving named basis directions. While the detailed process of compiling the computational graph into a CRAFT model is beyond this review's scope, in summary, the CRAFT compiler specifies how each operator applied to a given input type (numerical or categorical) maps to the parameters of an attention head and/or MLP. For operations like Map or Sequence Map, these compiled parameters primarily map values between inputs and outputs (see Figure 8), as the domain and range of each operation have been established earlier.

In our example program, the first operator is the select operation, producing a confusion matrix with inputs $Q$ and $K$ both equal to "hello" and using the equality predicate. The resulting confusion matrix from the attention head with parameters $W_{QK}$ is $Q \times W_{QK} \times K^\top$. In Figure 12, the diagonal is 1 for $x >= 2$ (matching tokens with themselves), while entries $(4, 5)$ and $(5, 4)$ are also 1 due to the two occurrences of 'l' in the input. After the Select operation, we move on to the selector width operation, where the SoftMax activation is applied, and the $W_{OV} \times V$ matrix selects the 'ones' column as the output. Here, SoftMax computes the sum of the original confusion matrix along the row axis in this context.

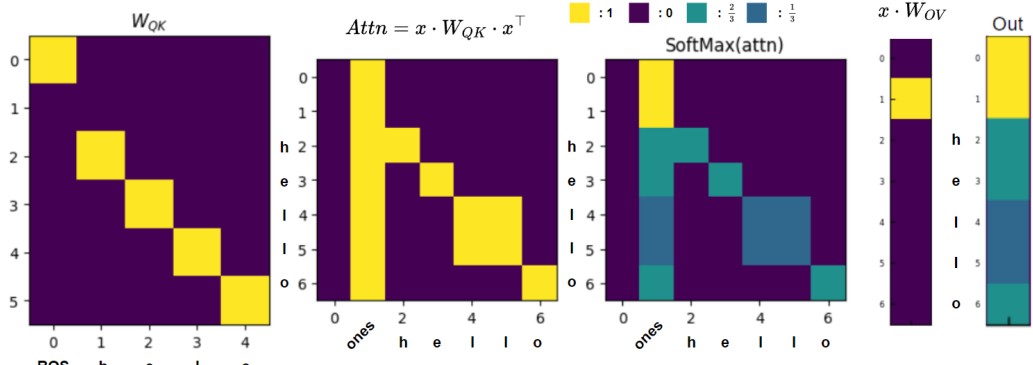

Figure 12: Select operation, using an attention head to compute the predicate 'equals' between the tokens, resulting in a confusion matrix. The Query, Key parameter is applied to the token inputs giving the 2nd figure. Applying softmax causes the ones column to act as an inverted accumulator, where $\frac{1}{2}$ corresponds to a token frequency of 1, and $\frac{1}{3}$ corresponds to a token frequency of 2. The Value parameter times the inputs, causes just the ones column with the inverse accumulated outputs to be kept

The outputs now contain a scalar encoding of the histogram values over our input tokens, however, we wish for them to be one hot encoded, which is the job of the MLP.

The first MLP layer matrix has a bar of 100's and below that a scale that exponentially decreases from -15 to -75. The result is the same scale multiplied by the attention outputs, such that the two rows corresponding to 'l' are 2x the rows corresponding to 'h', 'e' and 'o'.

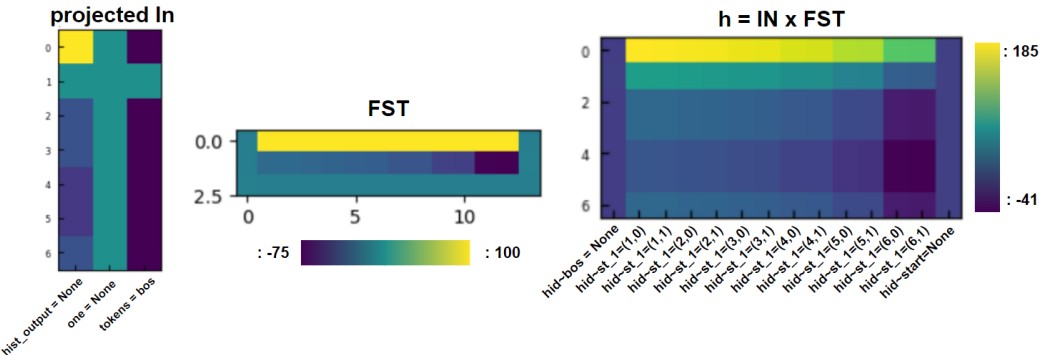

Figure 13: Inputs to first MLP layer on left, the first layer applies a gradient, resulting in the gradients in the ouptuts $h$

Next, we apply a ReLU activation to the outputs and then multiply by the second layer parameters, whose alternating checkerboard pattern cause the signals from incorrect indices to cancel leaving just a one-hot encoding of the frequencies of the input tokens.

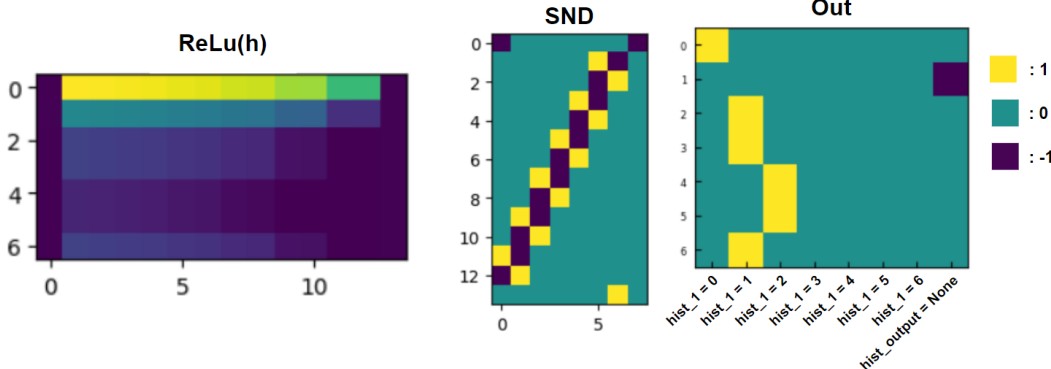

Figure 14: First ReLU is applied to the outputs of the first layer, then the alternating checkerboard pattern causes signals in the gradient to cancel out, resulting in the one-hot encoding of the token frequencies on the right

In summary, we've discussed the process of transforming a basic RASP program into a functional transformer program that accomplishes the same task. We've also examined the compiled parameters required to achieve this transformation. Additionally, another compiler step introduced by Tracr involves compiling the previously mentioned CRAFT parameters into a JAX transformer. This step is relatively straightforward and involves copying the parameters while padding them with zeros to ensure that all key, query, and value weight matrices have the same shape.

## C.1 Tracr Compilation

The CRAFT compiler in Tracr incorporates a technique for combining attention heads and MLPs within the same block efficiently. In Figure 18, the program branches into two separate computations, namely $Select \rightarrow SelectorWidth$ and $Select \rightarrow Aggregate$, each requiring an attention head and a 2-layer MLP. Since these computations are independent, they can run in parallel. Combining the attention heads is straightforward, resulting in a multi-head attention layer with two distinct attention heads, leading to doubled matrices ($W_{QK}$ and $W_{OV}$) width. Managing MLP layers is a bit more complex, but thanks to the residual stream's structure, each MLP writes to mutually exclusive residual stream sections. By introducing an additional projection matrix to align their outputs with the correct residual stream section, the MLP parameters can be concatenated. This projection matrix can then be multiplied into the second layer's parameters, resulting in a single two-layer MLP that handles both the selector width and aggregate operations and correctly writes the output to the respective regions of the residual stream.

## D  Inverse Tracr

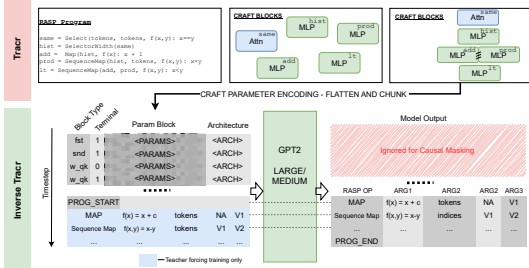

Figure 15: **Top:** Tracr (Lindner, Kramár, Rahtz, et al. 2023) is a method for compiling RASP code to equivalent transformer weights. **Bottom:** Our meta-model architecture for inverting Tracr. The meta-model conditions on the compiled weights ('Param blocks' on left) and autoregressively (teacher forcing input tokens on left) predicts the corresponding RASP code.

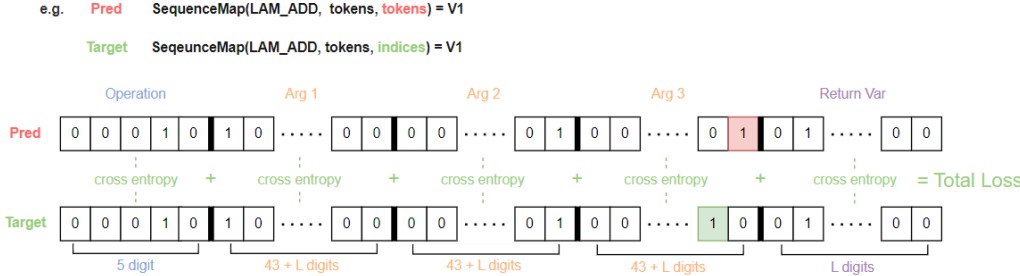

Figure 16: Segmented cross entropy loss between program lines - Above there are example predictions and targets where we are incorrectly predicting arg3 as tokens, below are their encoded representations with 5 segments corresponding to one hot encodings of the operation, 3 arguments and return variable. The cross entropy loss is then taken between each of these segments to measure the distance between the predicted and target sequences.

## D.1   Program Generation

```
1  initial_scope = {tokens, indices}
2  operations = []
3  for n in range(0, n-1):
4      op = sample_rasp_operator(scope, RASP_OPS)  #Sample a new
        ↪    function to add to the program
5      operations.append(op)
6
7  def sample_rasp_operator(scope, RASP_OPS):
8      op = sample(RASP_OPS)
9      switch op:
10         case Map:
11             lam = sample(Categoric_Lambda | Numeric_Lambda)
12             if lam is categoric:
13                 return Map(var(SOp), gen_const(CAT_OR_NUM), lam)
14             elif lam is Numeric:
15                 return Map(var(SOp), gen_const(NUM) + noise(),
                   ↪    lam)
16         case SequenceMap:
17             lam = sample(Numeric_Lambda)
18             v1, v2 = vars(2, SAME_TYPE)
19             return SequenceMap(v1, v2, lam)
20         case Select:
21             pred = sample(Predicate)
22             v1, v2 = vars(2, SAME_TYPE)
23             return Select(v1, v2, pred)
24         case Aggregate:
25             v1 = var(SELECT)
26             v2 = var(Numeric)
27             return Aggregate(v1, v2)
28         case SelectorWidth:
29             return SelectorWidth(var(SELECT))
30
```

Figure 17: Simplified RASP Program Generation Algorithm. `Var(X)` samples a variable of type X from the current scope. `vars(2, SAME_TYPE)` samples two variables of the same categoric/numeric type within the current scope.

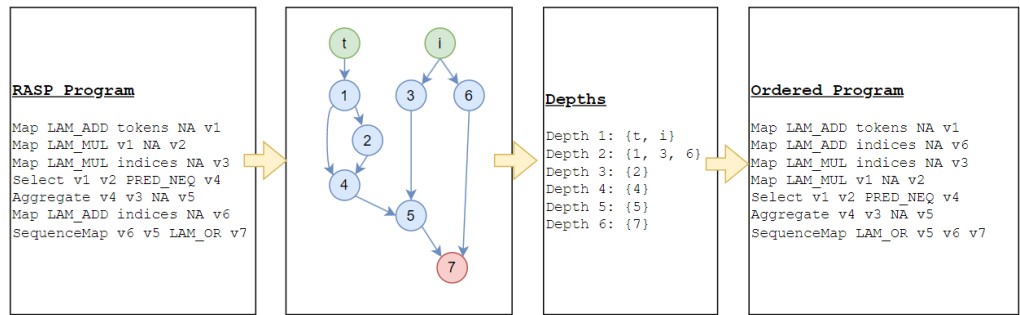

Figure 19: Computational Depth Program Ordering - given a program we construct the computational graph, compute the depths of each node in the graph w.r.t. the tokens and indices, allowing us to order the program by computational depth, breaking ties alphabetically

| RASP OP | Categoric Lambda | Numeric Lambda | Predicate |
|---|---|---|---|
| Map | $x < y$ | x + y | EQ |
| Sequence Map | $x <= y$ | x * y | FALSE |
| Select | $x > y$ | x - y | TRUE |
| Aggregate | $x >= y$ | x or y | GEQ |
| Selector Width | $x! = y$ | x and y | GT |
| | $x == y$ | | LEQ |
| | not x | | LT |
| | | | NEQ |

Table 2: Relevant primitives that the program generator samples from

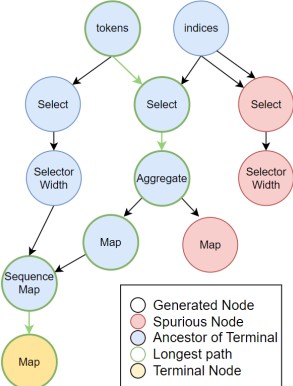

Figure 18: Pruning process and terminal node selection after sampling operations

## D.1.1   Example programs

```python
def example_program_1(tokens, indices):
    v1 = Select(PRED_NEQ, indices, tokens)
    v2 = SelectorWidth(v1)
    v3 = Select(PRED_LT, v2, v2)
    v4 = SelectorWidth(v3)
    v5 = Aggregate(v3, v4)
    v6 = SequenceMap(LAM_ADD, v2, v5)
    return Map(LAM_LE, v6)
```

Figure 20: A randomly sampled program generated using our algorithm, containing 2 attention heads and 2 map operations requiring MLP's

```python
def example_program_2(tokens, indices):
    v1 = Map(LAM_SUB, indices)
    v2 = SequenceMap(LAM_SUB, tokens, tokens)
    v3 = SequenceMap(LAM_MUL, v1, v1)
    v4 = Map(LAM_OR, v2)
    v5 = Select(PRED_TRUE, indices, v2)
    v6 = Aggregate(indices, v5)
    v7 = Select(PRED_LT, v3, v6)
    return Aggregate(v4, v7)
```

Figure 21: Another randomly sampled program generated using our algorithm, containing 2 attention heads and 4 map operations requiring MLPs

