# OpenReview forum: "Detecting Backdoors with Meta-Models"
_NeurIPS.cc/2023/Workshop/BUGS — NeurIPS 2023 BUGS Poster_

### Official Review · Reviewer_Rsgt · 2023-10-26
**This paper introduces a novel backdoor detection method via using meta-model.**

**Rating:** 7
**Confidence:** 3

**Review:**

### Summary

This paper introduces a novel backdoor detection method via using meta-models. Given the suspicious network, it is first flattened and divided into chunks of the same size. Each chunk is considered a token and is passed through a transformer decoder (meta-model) to classify. The author performs experiments on several settings, both iid and ood, to prove the effectiveness of the proposed method. The meta-model is also extended into network interpretability tasks and has shown it’s proficient

### Strengths

- The idea of using meta-models to detect backdoored model is novel to me and the technical details are solid
- The proposed method demonstrates superior performance compared to the baselines on different settings and poisoning methods.
- This paper also shows promising results on model interpretability tasks

### Weaknesses

- If the size of the architecture increases (e.g., to 10 million parameters), the number of tokens will increase accordingly, potentially requiring significant computational resources.

---

### Official Review · Reviewer_87MY · 2023-10-26
**Good promising work and well written paper that could benefit from another revision to flourish.**

**Rating:** 6
**Confidence:** 3

**Review:**

The paper proposes a novel approach to interpretability and backdoor detection - training transformers as meta-models. The paper's explanation of the models, the datasets, and evaluation metrics is articulate and concise. Experiments and results clearly show the contribution backdoor detection and techniques for general model understanding. Technically the paper is well written and sound. The figures support readability and drive home the author's explanations at multiple points in the paper. However, the paper has minor grammatical mistakes and some hard-to-understand language and could use a couple passes to refine the english. Further explanation of RASP and Tracr would help the readers unfamiliar with these tools. I say this because it is clearly integral to the paper and is not necessarily widely known to members of the community. The experiments, results and the setup could definitely use some more explanation in terms of what parameters were used for training and inference, motivation of individual experiments and the results could be better explained to clearly highlight the individual findings.

---

### Decision · Program_Chairs · 2023-10-28

**Decision:**

Accept (Poster)

**Comment:**

Thank you for submitting to our workshop. Both the reviewers recommend acceptance.